# Parameterized resetting model captures dose-dependent entrainment of the mouse circadian clock

Kosaku Masuda [1,2] ✉, Ryusuke Yoshimoto [1,2], Ruoshi Li [1,2], Takeshi Sakurai [1,2,3] & Arisa Hirano [1,2] ✉

The phase response curve (PRC) represents the time-dependent changes in circadian rhythm phase following internal or external stimuli. However, this time dependence complicates PRC measurement and quantification owing to its variable shape with changing stimulus intensity. Our previous work demonstrated that resetting a desynchronized circadian clock (singularity response, SR) simplifies the analysis by requiring only amplitude and phase parameters. In this study, we construct a comprehensive model for phase resetting in the mouse circadian clock by converting PRCs into SR parameters. We analyze single-cell PRCs and show that the SR amplitude parameters for different stimulus concentrations follow the Hill equation. Additionally, the model predicts the combined effects of multiple stimuli and pre-treatment (background) on phase response by simple addition or subtraction of individual SR parameters. Experimental validation using SR measurements in mouse cells and tissues confirms the model's accuracy. This study demonstrates that SRs facilitate PRC quantification and reveal simple rules governing phase resetting properties under various conditions using SR parameters.

The synchronization of the circadian clock is important for many organisms to adapt to environmental changes[1–3]. The phase response curve (PRC) summarizes the response of the circadian clock to stimuli at different phases and has traditionally been used to understand entrainment mechanisms[4–6]. Previous studies have measured PRCs for various stimuli, including combinations of stimuli, and reported nonlinear changes in their shape[7–12]. Although many PRC measurements have been attempted, comprehensive modeling for various conditions has not been achieved. One obstacle is the data-intensive nature of conventional PRC methods, requiring multiple measurements at four or more time points for a single PRC[7–10]. Single-cell imaging of desynchronized cell populations allows simultaneous measurement of many phase responses[11,12]. Using this method, the authors have shown nonlinear changes in PRC in response to complex stimuli such as a combination of glucocorticoid and forskolin at various concentrations[12].

This imaging-based approach opened possibilities for a comprehensive analysis of phase responses. However, analyzing large amounts of single-cell data while tracking each cell throughout several circadian cycles remains challenging. Another difficulty arises from the stimulus intensity-dependent changes in PRC topology[4–6,13]. Weak stimuli produce continuous PRCs (type-1), while strong stimuli result in discontinuous PRCs (type-0) with a point where the phase response reaches 12 h advance (or delay), causing discontinuity even with continuous stimulus intensity changes. This makes it challenging to compare PRCs on a common basis.

Our previous work demonstrated that the essential features of a PRC can be captured by the amplitude and phase of the singularity response (SR), which is the response of a desynchronized (singularity) state cell population in plants[14] and mammals[15]. This population comprises cells in distinct circadian phases; therefore, SR represents

[1]Institute of Medicine, University of Tsukuba, Tsukuba, Ibaraki, Japan. [2]International Institute for Integrative Sleep Medicine (WPI-IIIS), University of Tsukuba, Tsukuba, Ibaraki, Japan. [3]Life Science Center for Tsukuba Advanced Research Alliance, University of Tsukuba, Tsukuba, Ibaraki, Japan. ✉e-mail: kosaku.masuda@gmail.com; hirano.arisa.gt@u.tsukuba.ac.jp

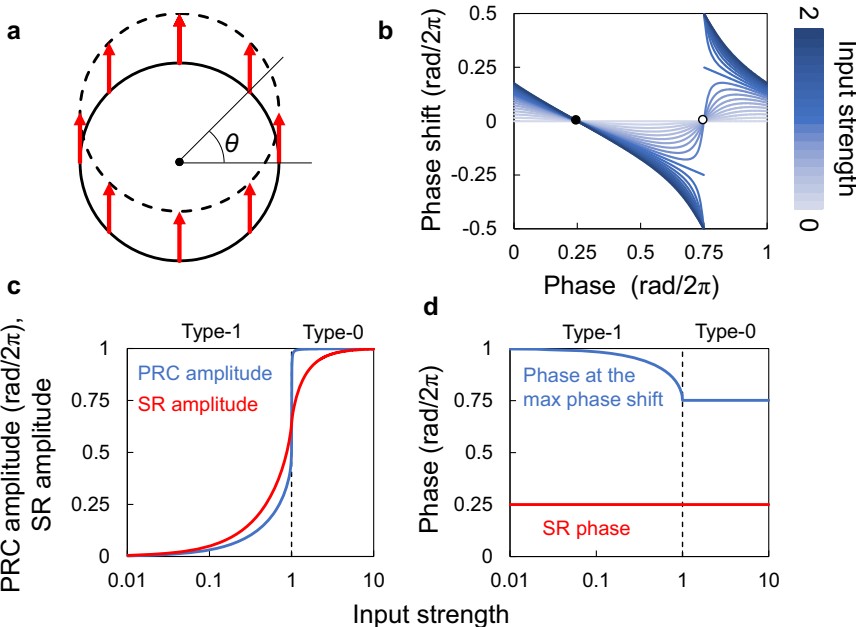

**Fig. 1 | PRC deformation and SR parameters. a** Phase response in a circular limit cycle model. The solid circle represents the pre-stimulus state, the dashed line represents the post-stimulus state, and the arrows represent stimulus-induced phase shifts. The input strength $F = 0.5$, and the input direction $\Phi = \pi/2$ radians. The angle from the center point represents the phase of the circadian rhythm, and the center point indicates a singular point. **b** Deformation of the phase response curve depends on the input strength. The input strength is represented by the length of the arrows in (**a**). The input direction ($\Phi$) is $\pi/2$ radians. Filled and open circles indicate stable and unstable states, respectively. **c** The maximum phase shifts induced by the PRC (blue line) and the SR amplitude (red line) are plotted against the input stimulus strength. **d** The phases at which the maximum responses occur in the PRC (blue line) and the SR phase (red line) are plotted against the input stimulus strength. PRC phase response curve, SR singularity response.

the average response of these cells in various circadian phases and the average response to these dispersed phases. The SR amplitude corresponds to the PRC amplitude, and the SR phase aligns with the stable point phase in the PRC. Consequently, the PRC can be mathematically reconstructed from these SR parameters.

SR measurements offer several advantages. They quantify the PRC with a single measurement on a desynchronized sample, eliminating the need for specific clock phases and cell-imaging equipment specialized for the analysis of phase response at a single-cell level. Additionally, SR parameters are independent of PRC type, enabling straightforward evaluation of parameters under various stimulating conditions. We previously indicated that SR can be measured for mammalian peripheral clocks, and that phase responses to the major entrainment factors, such as temperature and glucocorticoid, can be quantified using SR[15]. While the accuracy of our predicted PRC models based on SR has been well established in prior studies[14,15], the SR method has not been applied for high-throughput analysis of phase responses to diverse stimulating conditions, including dose-dependency and combination of distinct stimuli. Furthermore, few previous studies have thoroughly discussed the general property of entrainment needed to predict a PRC from known PRCs. In this study, we utilized the SR method to complex stimulating conditions to model the entrainment of the circadian clock. We calculated SR parameters from previously published single-cell-imaging data[12] to confirm the SR method's capability to capture dose-dependent changes in phase response. We also experimentally measured SR parameters to quantify PRCs for various stimuli, both individually and in combination, aiming to construct a comprehensive phase response model.

## Results
### Quantifying PRC using SR
To validate the use of SR parameters for quantitative PRC analysis, we used a simple phase response model[5]. This model assumed a stimulus

was delivered to a point on a circular limit cycle, with each point experiencing a parallel shift due to the stimulus (Fig. 1a). The phase response varied depending on the stimulus arrival phase, and the shape of the PRC varied depending on the input intensity (Fig. 1b). When the stimulus was weak, the PRC was type-1, where the PRC was continuous throughout all phases. Weak stimuli generate continuous type-1 PRCs, while stronger stimuli exceeding a central "singular point" induce discontinuous type-0 PRCs with a connected top and bottom.

Previous studies quantified PRC amplitude by measuring the phase response at a specific phase (often the point of maximum advance/delay) or by calculating the difference between maximum advance and delay for comparison across PRCs[11,12]. However, these methods were significantly affected by PRC type. The phase response abruptly changed near the unstable point, where it changed from negative to positive, but not at the stable point (from positive to negative). Notably, once a PRC becomes type-0, increasing stimulus strength has no further effect on its amplitude, which remains at the maximum value of 1 (Fig. 1c). In contrast, the SR amplitude, which indicates how strongly the stimulus synchronizes the rhythms, increases continuously, monotonically, and exponentially at low input intensity, asymptotically approaching a maximum value of 1 at high input strength. This behavior aligns with a Hill curve shape. In experimental PRC, if the number of phases at the stimulation is small, it may reduce the accuracy of SR parameter estimation (Supplementary Fig. 1). This error is also especially large at the transition from type-1 to type-0 PRCs.

Conventionally, the phase inducing maximum advance or delay in the PRC is used as a characteristic phase. However, this value can also vary with stimulus intensity (Fig. 1d). Conversely, the SR phase, indicating the phase at which the stimulus entrains the rhythms, was constant and independent of stimulus intensity for stimuli with a consistent direction (red arrows in Fig. 1a). However, the SR phase adjusts when the stimulus direction (angle of red arrows) changes

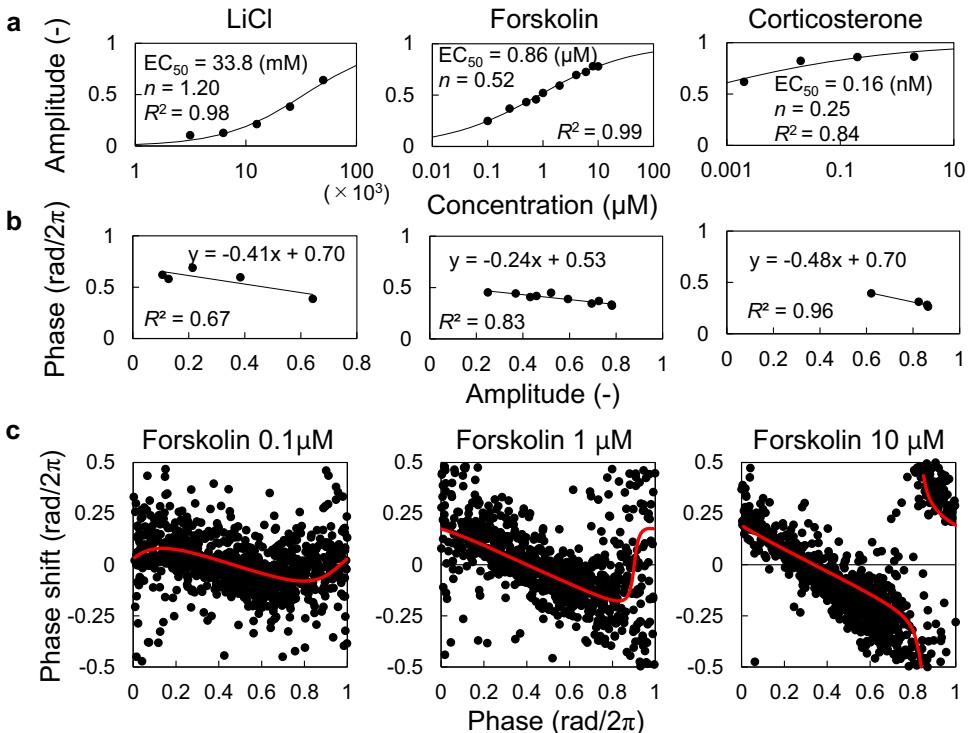

Fig. 2 | **Dose-response curves for SR parameters. a** The SR amplitude is plotted against the concentration of lithium chloride (LiCl), forskolin, and corticosterone. Each curve represents a fitted curve, and the corresponding parameters are shown in the illustration. **b** The relationship between SR amplitude and phase is shown for each stimulus. Each line represents a fitted line, and the corresponding approximate expressions are shown. **c** PRCs for forskolin at indicated concentrations (black points) and the estimated PRCs based on the SR parameters obtained from the dose-response curves (red curves). PRC phase response curve, SR singularity response.

while intensity remains constant (Supplementary Fig. 2). These results demonstrate the robustness of SR parameters in representing the strength of responsiveness and reset phases, regardless of the PRC type.

## Quantifying resetting properties with SR for various stimuli

Previous studies measured PRCs for a variety of stimuli at various concentrations by imaging the desynchronized cell population of NIH3T3 cells[11]. Using these PRCs, we calculated SR parameters and evaluated the resetting properties of the stimuli (Fig. 2a, b, Supplementary Fig. 3). For most stimuli, the SR amplitude showed either an exponential increase at low values, a linear increase in the middle range, or an asymptotic approach to a maximum of 1 at high values (Fig. 2a and Supplementary Fig. 3). As predicted in Fig. 1c, these are the components of the Hill equation, which is often used as a dose-response curve. Our previous study also showed that the SR amplitude for dexamethasone (DEX) follows Hill's equation[15]. Therefore, the dose-dependence of SR amplitude can be approximated by the Hill equation: $H(x) = x^n / (x^n + EC_{50}^n)$, where $x$ is the concentration. Notably, dose-dependent changes in SR amplitude can be described by only two parameters: the median effect concentration ($EC_{50}$) and the slope parameter ($n$). The SR phase varied linearly to the SR amplitude (Fig. 2b and Supplementary Fig. 3). Consequently, the entire phase response can be predicted using Hill's equations. These findings indicate that SR parameters (amplitude and phase) for various stimulus concentrations can be represented by two dose-response curves requiring only four parameters.

Our previous studies have shown that PRCs can be reconstructed using SR parameters[14,15]. Therefore, we can estimate the PRC at any stimulus concentration based on SR parameters derived from dose-response curves. Here, we fitted the Hill equations to the SR parameters for forskolin (FK), estimated SR parameters at each

concentration, and subsequently estimated the corresponding PRCs (Fig. 2c). Additionally, we directly analyzed the PRCs for FK using single-cell-imaging data. The results confirmed that the dose-dependent changes in the actual PRC for FK (black dots) were accurately reproduced by the SR parameters derived from Hill equation fitting (red lines).

## Predicting SR parameters for combined stimuli

Two models were considered for predicting the effects of combined stimuli on the phase response. The first model assumes that the stimulus-response is a vector on the limit cycle, with direction and strength represented by SR parameters (Fig. 1a). Therefore, the response to a mixture of stimuli can be considered the vector sum of individual SR parameters (Fig. 3a). We initially validated this model using data from a previous study that used single-cell imaging to measure PRCs[12]. We then compared the SR parameters obtained from the PRCs for each combination of DEX and phorbol 12-myristate 13-acetate (PMA), DEX and FK, and cobalt chloride ($CoCl_2$) and LiCl with the sum of SR parameters from individual stimuli. The experimental and predicted SR amplitudes exhibited good agreement when the predicted SR was <0.8 (Fig. 3b). However, the maximum experimental amplitude was approximately 0.8, whereas the predicted amplitude exceeded 1 (Fig. 3b). This suggests that the experimental amplitude asymptotically approaches a maximum value of 1. Thus, we capped predicted amplitudes exceeding 1 at 1. Regarding phase, the predicted values were slightly advanced 0.1 radians (rad/2π) compared with experimental values (Fig. 3c). As shown in Fig. 2, increasing stimulus concentration often leads to a delayed SR phase. Therefore, the combined effect of stimuli might exhibit a delay at higher concentrations.

Increasing the concentration of the resetting reagent can be interpreted as adding two different concentrations of the same

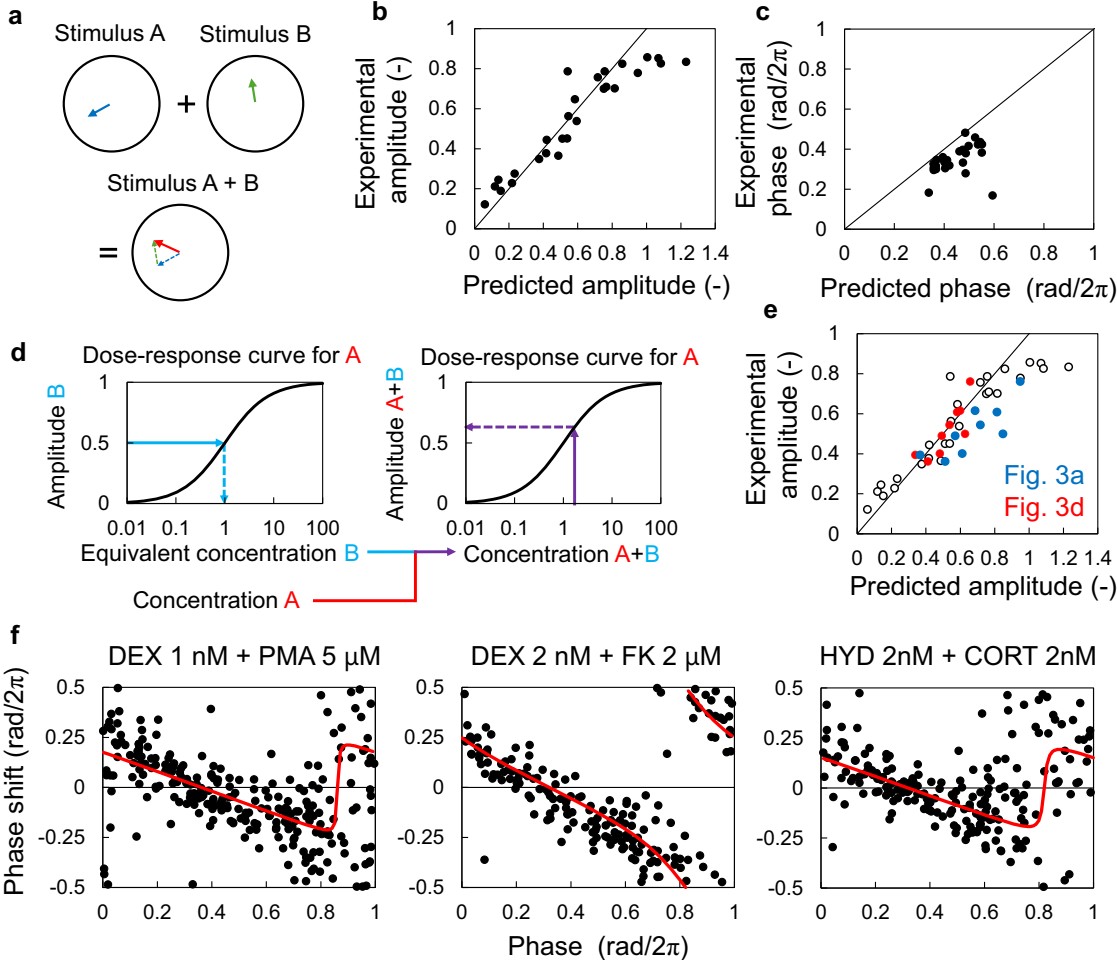

**Fig. 3 | Predicting SR parameters for combined stimuli. a** Prediction model for mixed stimuli based on SR parameters for each single stimulus. **b** Predicted SR amplitude of the combination of different types of stimuli: dexamethasone (DEX) and phorbol 12-myristate 13-acetate (PMA), DEX and forskolin (FK), and cobalt chloride and lithium chloride. **c** Predicted SR phase for the combination of different types of stimuli. **d** Prediction model for mixed stimuli based on the equivalent concentrations. **e** Prediction of the SR amplitude for the combination of the same types of stimuli (hydrocortisone (HYD) and corticosterone (CORT)). Red and blue points indicate the estimated amplitude using the sum of equivalent concentrations (**a**) and SR parameters (**d**), respectively. Blank circles indicate the results of the combination of different types of stimuli (**b**). **f** Measured and predicted PRCs for mixed stimuli (black points), and predicted PRCs based on the predicted SR parameters (red curves). PRC phase response curve, SR singularity response.

reagent. In this case, the increase in SR amplitude with increasing concentration deviates from the simple sum of individual SR amplitudes due to the Hill curve dependence. This is also expected to occur when two distinct stimuli are given simultaneously if they share the same signaling pathway for phase response, such as hydrocortisone (HYD) and corticosterone (CORT), which activate glucocorticoid receptors. Furthermore, each stimulus does not necessarily have the same effect at the same concentration because of its different binding affinities. Therefore, we obtained equivalent concentrations for one stimulus by comparing the SR amplitude of that stimulus with the dose-response curve of the other stimulus (Fig. 3d). The SR amplitude for a combination of stimuli can then be predicted by substituting the sum of the equivalent concentrations of each stimulus into the dose-response curve. Here, we obtained equivalent concentrations by comparing the SR amplitudes of CORT and the dose-response curve for HYD. Subsequently, the sum of equivalent concentrations was used in this dose-response curve for prediction. The SR amplitudes predicted using the sum of equivalent concentrations closely matched the experimental values (Fig. 3e). When we predicted the SR amplitudes from a simple sum of the SR parameters, as shown in Fig. 3a, the predicted amplitudes were larger than those obtained from experimental PRCs. Therefore, for stimuli targeting the same signaling

pathway, obtaining amplitude parameters through dose-response curves and equivalent concentrations is required (Fig. 3d). The SR phase can be estimated based on the linear relationship between amplitude and phase observed in Fig. 2b and Supplementary Fig. 3b (Supplementary Fig. 4). Based on these findings, we confirmed that estimated PRCs agreed with experimental results (Fig. 3f).

**Predicting SR parameters for background effects**

If the stimulus was administered to the cells twice, then the final response was assumed to be equal to that of the two stimuli administered simultaneously. In other words, the response to the second stimulus is predicted by subtracting the response to the first stimulus from the combined response. Considering the first stimulus as the background, which also indicates the level of resetting reagents in the medium, and the second as an additional stimulus, the background effect of the resetting reagents was estimated (Fig. 4a). This model predicts that the response to an additional stimulus is attenuated as the background concentration increases (Fig. 4b). However, a previous study administering the second stimulus 7.5 d after the first suggests a potential decrease in stimulus concentration in the medium or cellular sensitivity over time[12]. In this scenario, the observed phase response would be smaller than the sum of responses induced by the two stimuli (Fig. 4c).

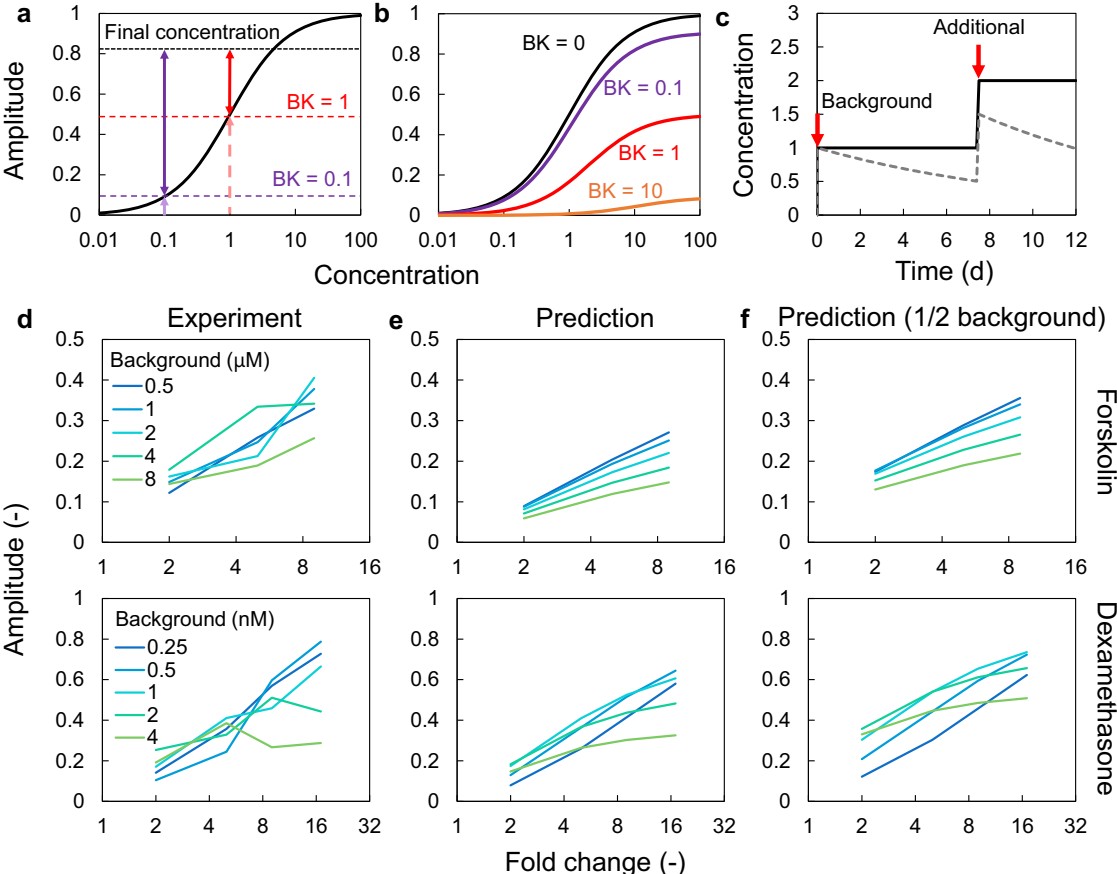

**Fig. 4 | Background effects on SR parameters. a** Prediction model of the background effect. Dashed arrows represent SR amplitude for background stimuli (BK). Solid arrows represent the SR amplitude induced by the additional stimulus. **b** Attenuation of SR amplitude depends on background concentration. **c** The decay of background concentration during the culture. Solid and dashed lines indicate the concentration of a stimulating chemical or molecules in the medium without and with decay, respectively. **d** SR amplitude for forskolin and dexamethasone with background effects as a function of fold change. **e**, **f** Model predictions of background effect without (**e**) and with (**f**) background decay. SR singularity response, BK background stimuli.

We employed these models to predict changes in responsiveness to background effects for FK and DEX, comparing the predicted results with the experimental results (Fig. 4d–f). In the experiment, the SR amplitude for FK increased proportionally to the fold change (ratio of the background to the final concentration), regardless of the absolute background concentration. However, the SR amplitude for DEX was saturated at high fold changes when the background concentration was high. We used previously established dose-response curves for each amplitude parameter of FK and DEX to predict the influence of background stimuli. We first assumed that the concentration in the medium did not change with the incubation time before the second administration. The predicted response for FK increased proportionally to the fold change, while DEX showed a smaller response compared with experimental values (Fig. 4e). The SR amplitude for DEX was proportional to the fold change in the DEX dose at low background concentrations; however, at background high concentrations, the response was saturated. This result was similar to the experimental results (Fig. 4d and Fig. 4e). As with single stimuli, the SR phase varied linearly with SR amplitude (Supplementary Fig. 5).

Next, we assumed that the concentration in the medium decreases with time and becomes half by the time of the second stimulation due to the metabolism of the compounds. The predicted values for FK were higher than those shown in Fig. 4e and closer to the experimental values compared with the constant concentration assumption (Fig. 4f). The dose-dependency of the SR amplitude for DEX did not significantly change compared with the previous condition (Fig. 4d–f). However, the predicted response exceeded the experimental values, likely due to not accounting for background stimulus decay. From these results, accounting for the changes in medium concentration or cellular sensitivity can improve the accuracy of background effect predictions. However, the rate of these changes may vary depending on the specific stimulus.

### Direct measurement of SR and model validation

While SR parameters were previously obtained from PRCs at the cellular level, our prior work demonstrated the feasibility of direct SR measurement from desynchronized cell populations without PRCs[14,15]. This has also been verified by numerical simulations (Supplementary Note 1, Supplementary Fig. 6). Here, we employed mouse embryonic fibroblasts (MEF) prepared from *PER2::LUC* knock-in mice[16] to quantify the phase resetting properties of various stimuli using direct SR measurements. Since cultured cells desynchronize spontaneously under constant conditions, we applied a stimulus 4 d after initiating the measurement to assess SR. The responsiveness to each stimulus was determined by comparing the SR amplitude with the control condition (Supplementary Fig. 7). Initially, SRs for a series of concentrations were measured for DEX, FK, and CORT (Fig. 5a). We confirmed that the SR amplitude followed the Hill equation and the phase varies linearly with amplitude for sufficiently strong responses (Fig. 5b, c), which is similar to the results in Fig. 2. However, the direct measurement of SR showed a maximum value of 0.6, which was smaller than the SR parameters obtained from the actual PRC. This may be because the amplitude of the *PER2::LUC* rhythm at the single-cell level is not necessarily 1. To address this, dose-response curves were normalized by dividing each value by the maximum value of the

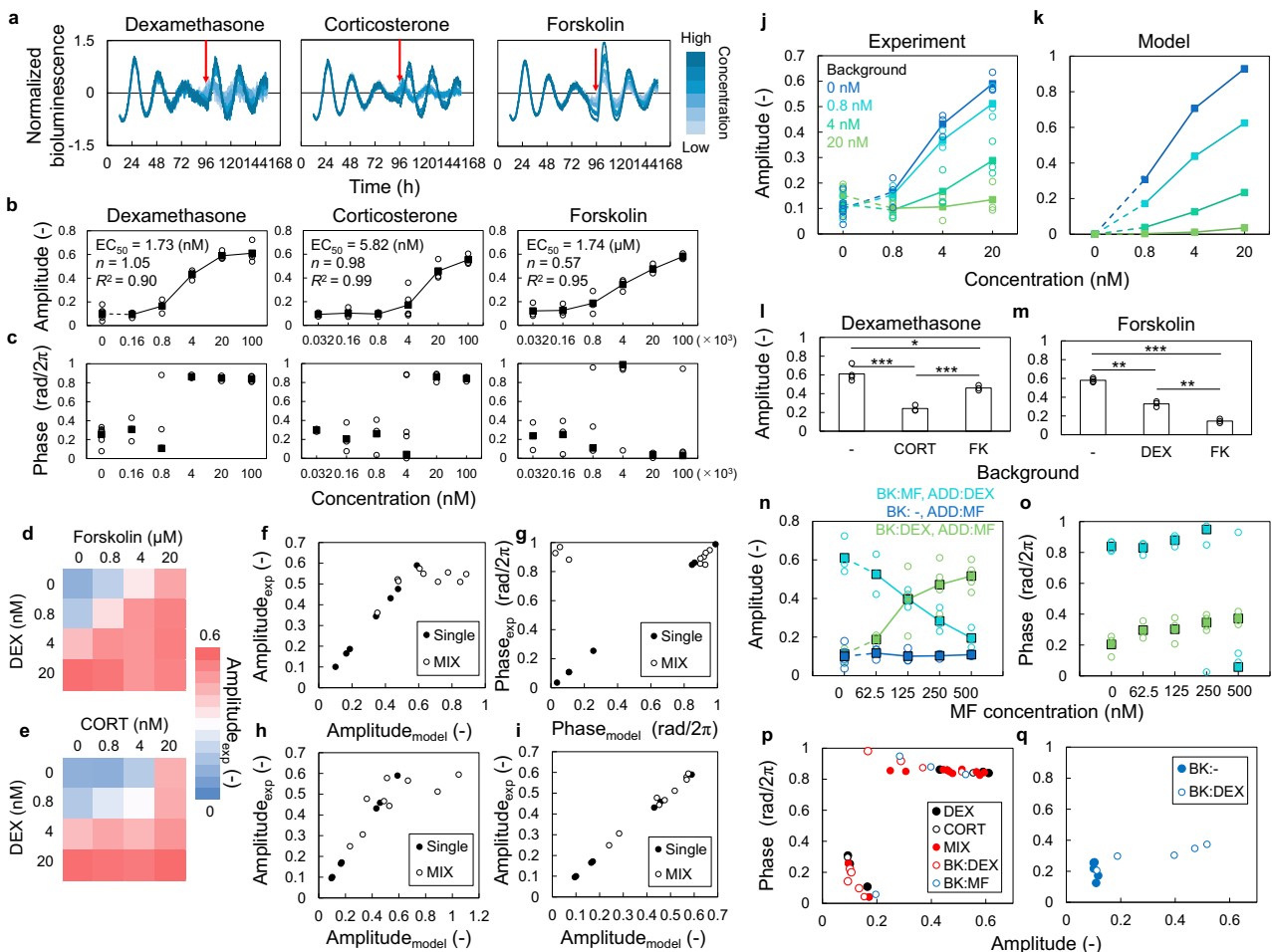

**Fig. 5 | Evaluation of SR parameters by direct measurement of SR in cultured cells. a** Normalized bioluminescence of *PER2::LUC* at SR measurement using dexamethasone (DEX), forskolin (FK), and corticosterone (CORT). The concentration of each line corresponds to that in (**b**) and (**c**). **b** Dose-response curve of SR amplitude. Each parameter was obtained by fitting it to the Hill equation. **c** Change in SR phase with different concentrations. The blank circle is individual data, and the filled square is the mean value. **d, e** Changes in SR amplitude for mixed stimuli of the combination of DEX and FK (**d**), and DEX and CORT (**e**). **f, g** SR amplitudes (**f**) and phase (**g**) for the combination of DEX and FK are estimated by the sum of SR parameters for each stimulus. **h, i** SR amplitudes for the combination of DEX and CORT are estimated by the sum of SR parameters (**h**) and equivalent concentration (**i**). Filled circles indicate the results of single stimuli, and blank circles show the results of mixed stimuli. **j, k** Change in SR amplitude for DEX with DEX background in experiment (**j**) and model (**k**). **l, m** Change in SR amplitude for 100 nM DEX (**l**) and 100 μM FK (**m**) with different background: no background, 100 nM DEX, 100 nM CORT, or 100 μM FK (*$p < 0.05$, **$p < 0.01$, ***$p < 0.001$; two-tailed *t*-test). Exact *p*-values are provided in Source Data. **n, o** Change in SR amplitude (**n**) and phase (**o**) for the combination of 100 nM DEX and mifepristone (MF). The blank circle is individual data, and the filled square is the mean value. **p, q** Relationship between SR phase and SR amplitude for DEX and CORT stimulation (**p**) and MF stimulation (**q**). "BK" indicates the background stimuli, and "ADD" indicates the additional stimuli. Each individual data indicates the results of independent experiments and experiments were performed three times or more. SR singularity response.

response, and the Hill equation was fitted to obtain the parameters. Since smaller $EC_{50}$ values for SR amplitude indicate stronger responses, DEX had a stronger resetting effect than CORT, while each SR phase was the same at high concentrations. The SR phases for DEX and FK were significantly different ($p < 0.05$ between 100 nM DEX and 100 μM FK, Watson–Williams test), likely reflecting the activation of distinct cellular signaling pathways by these stimuli. We also observed dose-dependent changes in SR amplitude for some stimuli that were reported to be involved in phase responses in previous studies (Supplementary Fig. 8)[11,12,17–19]. Hydrogen peroxide ($H_2O_2$) induced cell death at high concentrations and the amplitude peak appeared at moderate concentrations. In contrast, other stimuli showed dose-response curves conforming to the Hill equation. Comparing the parameters of each dose-response curve, the $EC_{50}$ showed a wide range of values from approximately 1 nM to 10 μM, while the *n* values displayed less variation (0.6–1, Supplementary Fig. 9).

We subsequently measured SRs for mixtures of DEX and FK or CORT (Fig. 5d–i). When the stimuli were mixed, the responses were stronger than those elicited by single administrations (Fig. 5d, e). We predicted the SR amplitudes of the combination of DEX and FK using the sum of individual SR parameters, as modeled in Fig. 3a. These predicted values generally agreed with the measured values when the amplitude was 0.6 or less (Fig. 5f). Notably, the measured value remained constant at 0.6 even for higher predicted amplitudes. Since this value represents saturation for single stimulus responses, we also assumed a maximum response of 0.6 for the combination. The predicted and measured phases generally agreed; however, the predicted values occasionally displayed advanced phases (Fig. 5g). For the DEX and CORT combination activating the same signaling pathway, the model using the sum of equivalent concentrations showed better estimation results for the SR amplitude than the model using the sum of SR parameters (Fig. 5h, i).

We further investigated changes in responsiveness to DEX in the presence of background effects. Cells were cultured with DEX for 4 d before subsequent DEX treatment. As anticipated from Fig. 4b, responses to additional stimuli were attenuated as the background

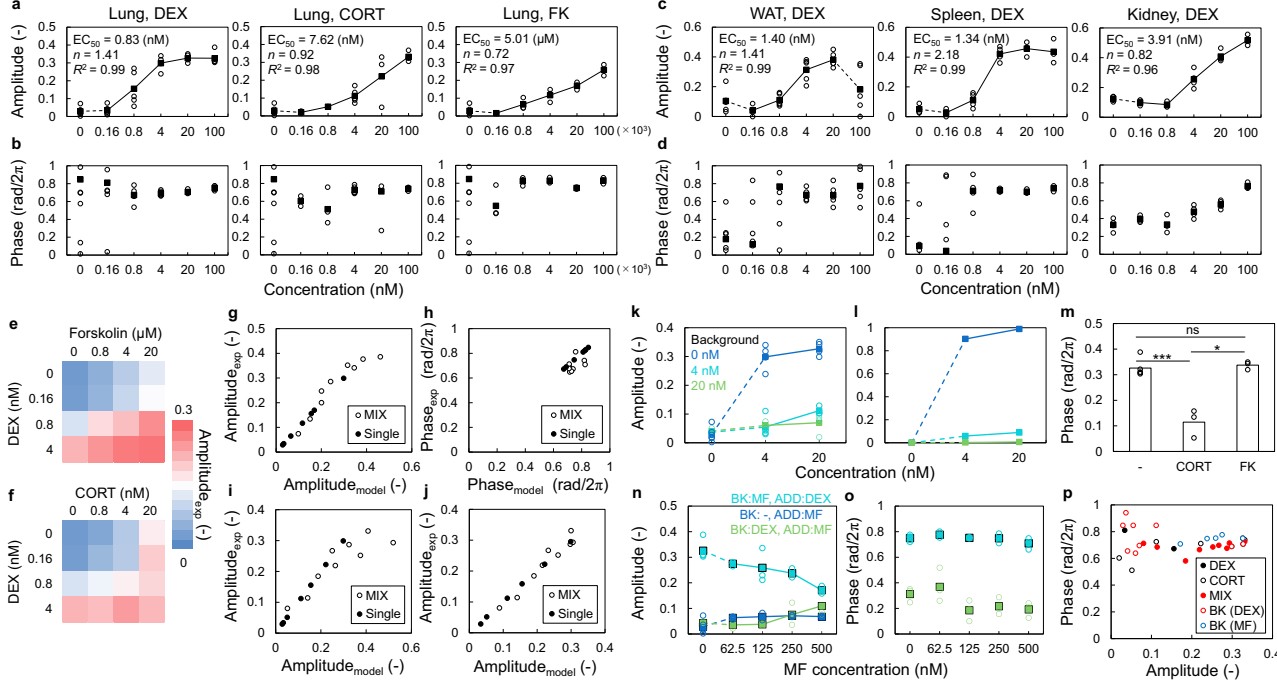

**Fig. 6 | Evaluation of SR parameters by direct measurement of SR in cultured tissues. a, b** Dose-response curve of SR amplitude (**a**) and phase (**b**) for dexamethasone (DEX), corticosterone (CORT), and forskolin (FK) in the lung. **c, d** Dose-response curve of SR amplitude (**c**) and SR phase (**d**) in white adipose tissue (WAT), spleen, and kidney. Each parameter was obtained by fitting using the Hill equation. Blank circles and filled squares indicate individual data and the mean values, respectively. In WAT, the dose-response curve was calculated excluding the value of 100 nM. **e, f** Changes in SR amplitude for the combination of DEX and FK (**e**) and DEX and CORT (**f**) in the lung. **g, h** SR amplitudes (**g**) and phase (**h**) for the combination of DEX and FK are estimated by summing SR parameters for each stimulus. **i, j** SR amplitudes for the combination of DEX and CORT are estimated by summing SR parameters (**i**) and equivalent concentration (**j**). Filled circles and blank circles

indicate the results of single stimuli and mixed stimuli, respectively. **k, l** Change in SR amplitude for DEX with DEX background (pre-treatment) experiment (**k**) and model (**l**). **m** Change in SR phase for 100 nM DEX with 100 nM CORT or 100 μM FK background (*$p < 0.05$, ***$p < 0.001$, ns: $p \geq 0.05$; two-tailed $t$-test). Exact $p$-values are provided in Source Data. **n, o** Change in SR amplitude (**n**) and phase (**o**) for the combination of DEX and mifepristone (MF). Blank circles and filled squares indicate individual data and the mean values, respectively. **p** Relationship between SR phase and amplitude for DEX and CORT stimulation. "BK" indicates the background stimuli, and "ADD" indicates the additional stimuli. Each individual data indicates the results of independent experiments and experiments were performed three times or more. SR singularity response.

concentration increased (Fig. 5j), aligning with model predictions (Fig. 5k). However, assuming a background concentration reduction to approximately one-third (1/3) during stimulation yielded results closer to the experimental values (Supplementary Fig. 10), suggesting potential mild DEX metabolism in the culture medium or cellular desensitization. High concentrations of CORT in the medium attenuated the response to DEX (Fig. 5l). Similarly, background FK treatment attenuated the response to an additional FK treatment (Fig. 5m). These findings confirmed that the resetting reagents in the medium attenuated the effect of the same additional stimulus. In contrast, while FK and DEX attenuated the response to DEX and FK, respectively (Fig. 5l, m), these background effects were less pronounced compared with those observed with the same type of stimulus, and no such interactions were reported in a previous study using combinations[12]. Therefore, a combination of different types of stimuli may produce minimal or no background effects.

Mifepristone (MF), an antagonist of progesterone and glucocorticoid receptors, attenuates the phase response to DEX[11,20]. We tested whether the inhibitory effect of MF on DEX could be quantified by SRs. As expected, the response to DEX was attenuated with increasing MF concentrations, confirming that the inhibitory effects of phase resetting can be quantified by SRs (Fig. 5n). Single-stimulus administration of MF elicited minimal phase response even at high concentrations. However, when added to a medium containing a high concentration of DEX, MF induced a significant phase response. The SR amplitude of MF with a DEX background corresponded to the magnitude of the DEX response attenuation observed with MF co-

administration. The SR phase for MF with a DEX background exhibited an opposite shift compared with DEX stimulation alone (Fig. 5o). This result is consistent with the principle that when two stimuli are presented at different times, the response to the second stimulus is equal to the final response minus the response to the first stimulus, as shown in Fig. 3.

The relationship between the SR phase and amplitude observed with DEX and CORT is summarized in Fig. 5p. Each data point was plotted on the same curve under single, mixed, and background conditions. For MF administration, the phase opposed the reset phase of DEX and CORT at sufficiently high amplitudes, consistent with its antagonistic properties. However, the phases were similar for weak stimuli (Fig. 5q). This suggests that the reset phase may converge from the phase in the control condition to the stimulus-specific reset phase as the stimulus becomes stronger.

## Measurement of dose-response curves and model validation in tissue culture

We previously demonstrated the feasibility of measuring SR parameters in both tissues and cell cultures[15]. Here, we measured the SRs and validated our prediction model for the combination of stimuli using tissue cultures from *PER2::LUC* mice. We applied a stimulus 6 d after initiating the measurement to measure SR. To determine the responsiveness to each stimulus, the SR amplitude for each stimulus was compared with that of the vehicle control (Supplementary Fig. 11). Initially, we measured the responses of lung slice cultures to DEX, FK, and CORT (Fig. 6a, b). For each stimulus, the amplitudes varied

following the Hill equation, and the phases were reset to a similar value. Furthermore, we measured the SRs to DEX in the spleen, white adipose tissue, and kidney slice cultures (Fig. 6c, d). The amplitude of each SR varied according to the Hill equation, except for the white adipose tissue, which showed low bioluminescence at high concentrations. The kidneys displayed a slightly larger $EC_{50}$ compared with other tissues and cultured MEFs, although the values remained relatively close. In the SR phase, particularly at 4 nM and 20 nM, each tissue showed a delayed phase compared with MEFs, with the most pronounced delay observed in the kidneys.

In our previous experiments, long-term measurements revealed a low-amplitude state of *PER2::LUC* rhythm in the SCN[21]. Additionally, the SCN rhythm was shown to reset in response to FK stimulation[22]. To investigate whether SR to FK could be measured using SCN slices, we applied an FK stimulus 7 d after initiating the measurement. The results demonstrated that a low-amplitude state also emerged in the SCN, and the rhythm was reset in response to FK stimulation (Supplementary Fig. 12a). The dose-response analysis of the reset amplitude revealed a peak at 1 μM, followed by a decrease at higher concentrations, which contrasted with the Hill curves observed in cultured cells and peripheral tissues (Supplementary Fig. 12b). Notably, the reset phases were consistent across concentrations and resembled those observed in the lung slice (Supplementary Fig. 12c). These results indicate that the SCN exhibits SR-like responses; however, further studies are required to comprehensively characterize the SR in the SCN.

Next, the effects of the mixed stimuli were evaluated in the lung tissues (Fig. 6e–j). As observed in cell culture experiments, combined stimuli caused stronger responses than single administrations (Fig. 6e, f). For the combination of FK and DEX, the sum of the SR parameters successfully estimated the effect of the mixed stimuli (Fig. 6g, h). The model employing equivalent quantities provided superior estimates for the DEX and CORT combination compared with the model using the sum of SR parameters (Fig. 6i, j).

Furthermore, we evaluated the background effects of DEX in lung tissue cultures (Fig. 6k). As anticipated, high background concentrations attenuated the response to subsequent DEX exposure. The prediction model for background effects based on the dose-response curve showed results closely aligned with the experimental values (Fig. 6l). The presence of CORT in the medium attenuated the response to DEX, whereas FK had no significant effect on DEX stimulation (Fig. 6m).

We also evaluated the effect of MF as an inhibitor of glucocorticoid receptors that antagonize the effect of DEX. In lung tissue, MF inhibited phase resetting by DEX, although the attenuating effect was less pronounced compared with MEFs (Fig. 6n). We also confirmed the inhibition of DEX-mediated signaling by adding MF to the medium-induced phase resetting, whereas MF itself did not cause a phase response. The amplitude of this inverse response was approximately equal to the value attenuated by DEX inhibition, and the resetting phase was approximately opposite to that of DEX (Fig. 6o), which was similarly observed in cell culture. The reset phases to DEX and CORT under various conditions (single stimulus, combination, DEX- or MF-background) converged to similar values when the stimulus was strong (Fig. 6p).

## Discussion

In conclusion, this study demonstrates that SR parameters (amplitude and phase) for various stimuli can be effectively described using the Hill equation. This implies that, compared with conventional methods requiring several measurements for a single PRC, the dose-response curves can predict PRCs for other stimulus concentrations. Notably, these curves require a similar number of measurements as the conventional method for a single PRC. Furthermore, we show that simple models based on SR parameters and their dose-response curves can predict changes in PRCs due to combined stimuli and background

effects. Notably, these models were reproduced not only in cell cultures but also in tissue cultures. In addition, SRs enabled quantification of the inhibitory effects on clock resetting signaling. Here, we predicted the phase response caused by a glucocorticoid receptor antagonist based on the dose-response curve for DEX, an activator of the receptor. Based on these findings, we conclude that PRC can be easily quantified using SR parameters, and the changes in SR parameters under various conditions follow predictable patterns.

Our results, along with previous studies, confirm that pretreatment with resetting reagents attenuates the response to subsequent stimuli in both cultured cells and tissues (Figs. 5j–m and 6k–m)[12]. This suggests that, in vivo, a high baseline concentration of resetting factors in the blood and interstitial fluid weakens the entrainment of circadian rhythms. Indeed, chronic stress and aging are associated with elevated baseline glucocorticoid levels and circadian rhythm disruption[23–26]. Under these conditions, the resetting function of glucocorticoids, which usually peaks just before the active phase, may be weaker due to increased baseline secretion compared with normal conditions. Therefore, not only the misalignment or perturbation of resetting factors but also their average concentrations may serve as indicators of circadian rhythm disorders.

Furthermore, we demonstrated that while the glucocorticoid receptor inhibitor-induced a phase response in the presence of high DEX concentrations, it did not induce a phase shift on its own (Figs. 5n and 6n). This indicates that both increases and decreases in resetting factor concentration can cause a phase response under physiological conditions. Therefore, the inhibition of resetting factors may be a useful tool in circadian rhythm therapy, particularly when the concentrations of resetting factors are high and responsiveness is diminished. For example, some people have slow melatonin metabolism, which causes a loss of response to melatonin treatment; therefore, melatonin receptor antagonists may improve the circadian rhythms in such persons[27,28]. However, applying this model requires further validation beyond the DEX and MF combination explored here. Testing the effects with different cellular signaling pathways is necessary. Moreover, because MF is a competitive antagonist, investigating the effects of non-competitive antagonists is warranted.

This study demonstrates the feasibility of predicting phase responses under various conditions. This suggests the possibility of designing phase resets to achieve a targeted clock phase by combining stimuli. For example, DEX resets the clock around CT9 and forskolin resets the clock around CT13; however, their combination could reset the clock to an intermediate phase (Supplementary Fig. 13a). Notably, in MEFs, few stimuli reset the clock to a phase from CT15 to CT2. Previous studies have also reported similar nonuniformity in the reset phase[7,11,15]. Therefore, more extensive screening to identify more types of resetting stimuli may be necessary to control circadian rhythms more flexibly. In addition, the delayed resetting phases observed in lungs compared to MEFs highlight cell- and tissue-specific phase resetting characteristics (Supplementary Fig. 13b). The peak time of clock gene expression varies among different tissues in vivo[16,29,30]. Consequently, to control rhythm in vivo, we must consider the differences in phase responses between cells and tissues.

We have shown that SR is useful for quantifying the entrainment properties of the circadian clock under a variety of conditions. However, there are several limitations to the application of this SR-based method. First, although we did not consider the deviations from the limit cycle in the experiment if relaxation to the limit cycle takes longer time, there may be some error in SR amplitude prediction. If the relaxation to the limit cycle is fast, the error can be reduced by normalizing at the maximum response, as we did for the present experimental data (Supplementary Fig. 6). If the relaxation is extremely slow, as in the harmonic oscillators, the reset amplitude would continue to increase with stronger stimuli, which may be distinguishable from the present results. It is also possible that the predicted PRC may not be

able to explain the entrainment with the periodic environment (i.e. periodic light pulse), since the effects of deviations from the limit cycle may become more complex. However, such effects are likely to occur with particularly strong stimuli, so the present results quantifying the stimulus intensity are still important for understanding how the circadian clock synchronizes with various environments. Next, we considered a very simple PRC model in the present study (Fig. 1). From previous studies[8–12], the PRCs in cultured cells and tissues are usually sinusoidal when the stimulus is weak and linear when the stimulus is strong as shown in Fig. 1b. Therefore, we constructed a model for prediction of PRCs from SR parameters based on these typical PRC shapes (Figs. 2 and 3). However, PRC is measured not only in circadian rhythms but in other oscillating systems, such as neuronal firing, which may show a different shape from the PRC of circadian rhythms. Even though, since SR is a measure of how much and in which phase the rhythm of the desynchronized oscillators is concentrated by a stimulus, SR can be commonly observed regardless of the shape of the PRC. For example, in the integrate-and-fire models, the firing pattern only shows phase advances[31]. In such cases, the model we assumed in Fig. 1 may not correctly predict PRC. Therefore, for systems other than circadian clocks, the quantification of PRC using SR is still useful, but it would be necessary to assume a shape of PRC corresponding to that system to predict PRC from SR parameters. In addition, circadian rhythms are often not spontaneously desynchronized at the suprachiasmatic nucleus (SCN) or individual level. However, in our experiment, spontaneous decay was also observed in the SCN circadian rhythm, and SR-like responses to forskolin could be measured (Supplementary Fig. 12). Desynchronization of SCN is also induced by several treatments, such as tetrodotoxin, an inhibitor of voltage-gated sodium channels, MDL-12330A, an inhibitor of adenylate cyclase, and constant light conditions[9,21,32]. The dose-response curve of SR amplitude for forskolin in the SCN was not Hill curve-like, but at least for the single-cell, bulk, and peripheral tissue slices, the changes in SR parameters under various conditions can be represented by the same model based on the dose-response curve (Figs. 5 and 6). Future studies will show whether the SR-based models can express the entrainment properties in SCN and individual rhythms.

In this study, we proposed an SR-based model to predict the phase response of circadian rhythms under various conditions. While the mixing and background effects of combined stimuli can differ significantly depending on whether they share a common signaling pathway, this is not always a defining factor. One approach to determine this was to measure the SR of each stimulus with the background of the other. If the amplitude is significantly attenuated, they likely share a common signaling pathway; if it is unchanged, they likely induce phase shifts via different mechanisms. Thus, the proposed SR method can be used to assess the phase resetting mechanism. However, since combinations of different types of stimuli can also cause amplitude attenuation, further investigation is needed to refine the criteria for this determination. Although SR measurements were performed on mouse cells and peripheral tissues in this study, circadian rhythms in the SCN and other tissues with cellular networks might exhibit more complex responses to combined stimuli. Additionally, we do not know whether the proposed model applies to species other than mice. PRC has been measured in many species, including higher plants and unicellular organisms, and SR has been confirmed to be measurable in plants[14,33,34]. Therefore, we need to validate the SR-based quantification and modeling of PRC in various organisms, which may reveal the universal mechanisms by which organisms synchronize their internal and external rhythms.

## Methods

### Animals
All experiments involving animals were approved by the Animal Experiment and Use Committee of the University of Tsukuba and adhered to the National Institutes of Health (NIH) guidelines. Animals were housed under a 12-h light/dark cycle (12:12 LD) at a controlled temperature of $23.5 \pm 1.0\,°C$ and humidity of $50.0 \pm 10.0\,\%$. Food and water were provided ad libitum.

### Bioluminescence analysis using *PER2::LUC* MEFs
We employed *PER2::LUC* MEFs derived from *PER2::LUC* knock-in mice established in a previous study[16]. MEFs were cultivated in high-glucose Dulbecco's Modified Eagle Medium (DMEM) supplemented with L-glutamine, phenol red, and sodium pyruvate (catalog no. 043-30085, FUJIFILM Wako Pure Chemical, Osaka, Japan), 10% fetal bovine serum (FBS; Merck KGaA, Darmstadt, Germany), 100 units/mL penicillin, and 100 µg/mL streptomycin (catalog no. 09367-34; NACALAI TESQUE, Kyoto, Japan) at 37 °C under 5% $CO_2$. For bioluminescence recording, phenol red-free DMEM (Merck KGaA, Darmstadt, Germany; catalog no. D2902) containing 3.5 mg/mL D-glucose and 10 mM HEPES (adjusted to pH 7.0) was supplemented with 10% FBS, 100 units/mL penicillin, 100 µg/mL streptomycin, and 0.1 mM D-luciferin potassium salt (FUJIFILM Wako Pure Chemical, Osaka, Japan; catalog no. 126-05116) just before use. MEFs were plated in 24-well plates with 500 µL of culture medium 1 d before the measurement. The medium was then replaced with 500 µL of recording medium immediately before the measurement. Bioluminescence was monitored for 1 wk at intervals of 10 min using an automated monitoring device, Kronos HT (ATTO, Tokyo, Japan). In each experiment, stimuli were applied 96 h after the start of the measurement, and responses were measured over the following 3 d. For chemical stimulations, 50 µL of forskolin or 5 µL of each other agent diluted in phosphate-buffered saline (PBS) was added. The concentrations presented in the text represent the final concentrations. SR measurements were performed at least three times for each stimulus.

### Bioluminescence analysis using slice culture
To measure peripheral tissue slice culture, we used the lung, white adipose tissue, spleen, and kidney obtained from male homozygous *PER2::LUC* knock-in mice (C57BL/6 J genetic background, 3–8 months old). For SCN slice culture, we used male or female homozygous *PER2::LUC* knock-in mice (C57BL/6 J genetic background, 7–24 weeks old). Tissue slices were cut from mice at Zeitgeber time (ZT) 4 to ZT5. Each peripheral tissue piece was placed on a cell culture insert (Catalog No. PICM01250, Merck KGaA, Darmstadt, Germany) in a 24-well plate containing 500 µL of the recording medium (phenol red-free DMEM; Catalog No. D2902, Merck KGaA, Darmstadt, Germany). For SCN slice culturing, mouse brains were rapidly removed and placed in chilled 1× Hanks' balanced salt solution (Catalog No. 14025–092, Thermo Fisher Scientific, Waltham, MA, USA) on ice. Slices were prepared with 200 µm thickness using a vibratome (VT1200, Leica Biosystems, Nußloch, Germany), and a triangular area around 2 mm² including the SCN was cut out from each section. The collected piece was placed on a cell culture insert (Catalog No. PICM0RG50, Merck KGaA, Darmstadt, Germany) in a 35 mm tissue culture dish containing 1500 µL of the recording medium. The recording medium composition differed for kidney and SCN slices (phenol red-free DMEM with 2% B-27 supplement (Catalog No. 17504–044, Merck KGaA, Darmstadt, Germany), 3.5 mg/mL D-glucose, 10 mM HEPES, 35 mg/L NaHCO3 (Catalog No. 197–01302, FUJIFILM Wako Pure Chemical Corporation, Osaka, Japan), 100 units/mL penicillin, 100 µg/mL streptomycin (Catalog No. 15140-122, Merck KGaA, Darmstadt, Germany), and 0.2 mM D-luciferin potassium salt (Catalog No. 126-05116, FUJIFILM Wako Pure Chemical Corporation, Osaka, Japan) for kidney and 0.5 mM for SCN, pH 7.0) and other tissues (10% FBS, 3.5 mg/mL D-glucose, 10 mM HEPES, 100 units/mL penicillin, 100 µg/mL streptomycin, and 0.1 mM D-luciferin potassium salt, pH 7.0). Bioluminescence was monitored using the Kronos HT device for peripheral tissues and Kronos Dio (ATTO, Tokyo, Japan) for SCN culture, and stimulation was applied 6 days for peripheral

tissues and 7 days for SCN after initiating the measurement. SR measurements for each stimulus and tissue were performed at two times for SCN and at least three times for peripheral tissues.

## PRC model based on the circular limit cycle

A circadian clock can be modeled as a circular limit cycle with an amplitude of 1 (Fig. 1a). Suppose that the phase of the circadian clock before stimulation is $\theta_1$ and the displacement by stimulation is represented by $Fe^{i\Phi}$. Here, $F$ is the stimulus strength, $\Phi$ is the direction of the stimulus, and $i$ is the imaginary unit. In this scenario, the phase after stimulation $\theta_2$ is represented by:

$$\theta_2 = \arg\left\{e^{i\theta_1} + Fe^{i\Phi}\right\}. \tag{1}$$

If $e^{i\theta_1} + Fe^{i\Phi} = 0$, i.e., $F = 1$ and $\Phi = -\theta_1$, the circadian clock is at a singularity state with no defined phase. The phase response curve $g(\theta_1)$ is defined as:

$$g(\theta_1) = \theta_2 - \theta_1 \tag{2}$$

Since the singular point is exceeded at $F = 1$, $g(\theta_1)$ is a type-1 PRC when $F < 1$ and a type-0 PRC when $F > 1$.

## Calculation of SR parameters

**When obtained from the PRC.** Based on a previous study[15], the SR parameter is defined as follows:

$$Re^{i\Theta} = \frac{1}{2\pi} \int_0^{2\pi} e^{i\{\theta + g(\theta)\}} d\theta, \tag{3}$$

where $R$ is the SR amplitude, and $\Theta$ is the SR phase.

**When obtained from cell-level PRC data.** The cell-level PRC data used in this study were measured with an approximately uniform distribution of cell phases upon stimulation[11,12]. Therefore, the SR parameters were obtained from the PRC data using the following approximate formula:

$$Re^{i\Theta} = \frac{1}{N} \sum_{j=1}^{N} e^{i\{\theta_j + g(\theta_j)\}}, \tag{4}$$

where $j$ indicates the data of $j$th cell, and $N$ is the total cell number.

**When obtained from SR measurements.** Our previous studies showed that resetting the amplitude and phase of the circadian rhythm population by stimulation in a desynchronized state aligns with Eq. (3)[15]. Therefore, we obtained the SR parameters from the experimental values using the same method as in previous studies. First, luminescence was normalized using the following equations:

$$L_j = \frac{l_j - \bar{l}_j}{\bar{l}_j}, \tag{5}$$

$$\bar{l}_j = \frac{1}{2n+1} \sum_{k=0}^{2n} l_{j+k-n}. \tag{6}$$

Here, $l_j$ is the bioluminescence at $j$th time point, $\bar{l}_j$ is the moving average with a 24-h window, $L_j$ is the normalized bioluminescence, and $n$ is the number of data points within 12 h ($n = 72$). The phase and amplitude were then obtained by performing cosine fitting to the post-stimulus rhythm ($t = 24$–$48$ h after the stimulation for culture cells and peripheral tissues and $t = 24$–$96$ h for SCN). Cosine fitting was performed using the least-squares method with the Python library (lmfit; https://lmfit.github.io/lmfit-py/).

## Estimation of PRC using SR parameters

Assuming the phase response to the stimulation follows Eq. (1), the SR parameters are obtained as:

$$Re^{i\Theta} = \frac{1}{2\pi} \int_0^{2\pi} e^{i \arg\{e^{i\theta} + Fe^{i\Phi}\}} d\theta. \tag{7}$$

Here, the value of $R$ does not change when the value of $\Phi$ changes. $R$ increased continuously and monotonically before and after $F = 1$ when the PRC type changed (Fig. 1c). The larger $F$ is, the closer each post-stimulus phase approaches $\Phi$, so as $F$ increases, $R$ also approaches 1 asymptotically. Therefore, $F$ and $R$ correspond to each other. The PRC can be estimated by determining $F$ for the experimentally determined $R$ and substituting it into Eq. (1). In this study, the value of $F$ was optimized using the gradient descent method to minimize the squared error between $R$ from the experiments and Eq. (7). This calculation involved approximating Eq. (7) by the mean value of the integrand at $\theta_j = 2\pi j/100$ ($j = 0,\ldots,99$), and performing optimizations using the solver function of Microsoft Excel (version 2403 build 16.0.17425.20124).

## Calculation of dose-response curve

The Hill equation is expressed by the following equation:

$$H(x) = \frac{x^n}{x^n + EC_{50}^n}, \tag{8}$$

where $x$ is the concentration, $EC_{50}$ is the half-maximal effective concentration and $n$ is the slope parameter. The parameters ($EC_{50}$ and $n$) were optimized using the gradient descent method to minimize the squared error between the fitted curve and the experimental values. Calculations were performed using the solver function in Microsoft Excel (version 2403, build 16.0.17425.20124). For each approximate curve, the coefficient of determination $R^2$ was defined by the following equation

$$R^2 = 1 - \frac{\sum_{i=1}^{N}(y_i - f_i)^2}{\sum_{i=1}^{N}(y_i - \bar{y})^2}, \tag{9}$$

where $y_i$ is the measured value, $f_i$ is the estimated value and $\bar{y}$ is the average value of $y_i$.

## Estimation of the effect of combined stimuli

**For combinations of different types of stimuli.** Let SR parameters for stimuli A and B be $R_A e^{i\Theta_A}$ and $R_B e^{i\Theta_B}$, respectively. The SR parameters for the mixture of A and B, $R_{A+B} e^{i\Theta_{A+B}}$, are estimated by the sum of each SR parameter as:

$$R_{A+B} e^{i\Theta_{A+B}} = R_A e^{i\Theta_A} + R_B e^{i\Theta_B}. \tag{10}$$

If $R_{A+B} > 1$, we can estimate $R_{A+B}$ as 1 instead.

**For combinations of the same types of stimuli.** Let the dose-response curve for a given stimulus be $H(x) = x^n/(x^n + EC_{50}^n)$. When the SR amplitude of the same type of stimulus is $R$, its equivalent concentration is:

$$x_{eq} = H^{-1}(R) = EC_{50}\left(\frac{R}{1-R}\right)^{\frac{1}{n}}. \tag{11}$$

If the equivalent concentrations of two stimuli of the same type are $x_{eq,A}$ and $x_{eq,B}$, then the SR amplitude of the mixed stimulus can be

predicted as:

$$R_{A+B} = H\left(x_{eq,A} + x_{eq,B}\right). \tag{12}$$

The SR phase of the mixed stimulus can be predicted by substituting the predicted SR amplitude of the mixed stimulus into the relationship between the SR amplitude and SR phase, which was obtained from the results of the single stimulus, as shown in Fig. 5p and Fig. 6p.

## Estimation of the background effect

When a stimulus is applied twice consecutively, the final response is considered the sum of the responses to the first and second stimuli. Therefore, if the response to the first and final stimuli is known, the response to the second stimulus can be predicted from the difference between them. Here, the first and second stimuli were considered background and additional stimuli, respectively. Let the background concentration be $x_{BK}$ and the concentration of the additional stimulus be $x_{ADD}$. If the two stimuli are of the same type, the sum of the two stimuli is estimated using Eq. (12). The SR amplitude for the additional stimulus $R_{ADD}$ was obtained from the SR amplitudes for the background $R_{BK}$ and final response $R_{BK+ADD}$ as follows:

$$R_{ADD} = R_{BK+ADD} - R_{BK} = H(x_{BK} + x_{ADD}) - H(x_{BK}). \tag{13}$$

The SR phase can be predicted by substituting the predicted SR amplitude into the relationship between SR amplitude and SR phase, obtained from the results of the single stimulus, as shown in Fig. 5p and Fig. 6p.

For combinations of different stimuli (as described in Eq. 10), the response to the additional stimulus is always equal to the difference between the final response and the response to the first (background) stimulus. Therefore, the background had no effect on the additional stimuli of a different type.

If the additional stimulus is an inhibitor, $R_{BK+ADD}$ becomes smaller than $R_{BK}$, thus, $R_{ADD} < 0$. In this case, the SR amplitude for the additional stimulus can be estimated as $|R_{ADD}|$, and the SR phase as the opposite of the reset phase of the inhibitory target.

## Simulations using the Stuart-Landau model

The Stuart-Landau model is represented by the following equation[35],

$$\frac{dZ}{dt} = (\sigma_r + i\sigma_i)Z - \frac{(l_r + il_i)}{2}Z\left|Z^2\right|, \tag{14}$$

where $i$ is an imaginary unit and $Z$ is a complex variable. Here, we put $l_i = 0$, and Eq. (14) is rewritten as

$$\frac{dR}{dt} = \alpha\left(A - R^2\right)R, \tag{15}$$

$$\frac{d\theta}{dt} = \sigma_i. \tag{16}$$

$R = |Z|$ is the distance from the origin, $\theta = \arg Z$ is the angle. $\alpha = l_r/2$ is a parameter representing the speed of recovery to the limit cycle. $A = 2\sigma_r/l_r$ is the amplitude of the limit cycle and $\sigma_i$ is the frequency of the oscillator. We set $A = 1$ and $\sigma_i = 2\pi/24$. In this model, the point on the limit cycle is represented by $Ae^{i\theta}$. Consider that each point on the limit cycle moves to $Ae^{i\theta} + Fe^{i\Phi}$ by a stimulus. For 100 of oscillators, the initial phase was set to $\theta_j = 2\pi j/100$ ($j = 0,...,99$), and the collective output was calculated as the average value of $R_j \cos\theta_j$ for each oscillator's amplitude $R_j$ and phase $\theta_j$. SR amplitude and phase were obtained by cosine fitting to the rhythm at $t = 24$–48 after the stimulation.

## Reporting summary

Further information on research design is available in the Nature Portfolio Reporting Summary linked to this article.

## Data availability

Source Data are provided with this paper. All data needed to evaluate the conclusions of this study are presented in the manuscript, Supplementary Information, and/or Source Data. The raw bioluminescence data generated in this study are provided in the Source Data file.

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

## Acknowledgements

We are grateful to Dr. H. Yoshitane for providing the *PER2*::*LUC* MEFs. This study was funded by the Japan Society for the Promotion of Science (Nos. 22KJ0377 and 22K15157 to K.M.), the Moonshot Research Development Program (No. JP21zf0127003 to A.H. and JP21zf0127005 to T.S.), and AMED-PRIME (No. JP22gm6410030 to A.H.) provided by the Japan Agency for Medical Research and Development (AMED).

## Author contributions

K.M. and A.H. designed this study. K.M., R.Y., and R.L. performed the experiments. K.M. analyzed the data. K.M., T.S., and A.H. drafted the manuscript. T.S. and A.H. supervised the study. All authors discussed the results and implications and commented on the manuscript.

## Competing interests

The authors declare no competing interests.
