## [Transparent Peer Review file · Nature Communications]

Parameterized resetting model captures dose-dependent entrainment of the mouse circadian clock

Corresponding Author: Dr Arisa Hirano

Version 0:

Reviewer comments:

Reviewer #1

(Remarks to the Author)

This article is another in a sequence of articles in which the authors expound the benefits of the singularity response in circadian biology. In this installment, the dose-response is studied. The results show that the singularity response is an effective and efficient method to measure phase-response curves for a range of stimuli strengths. Remarkably, the singularity response is effective for multiple stimuli, as well as in both cultured cells and tissues.

My only significant suggestion is to include a summary of the prior work in the introduction. Detail the author's prior results on the singularity response and put the new results in this article in a proper context.

minor comments:

- 0) Figure 1 illuminates the approach well. Is there a good reason to use a different input direction $\Phi = 0$ vs $\Phi = 0.25$ in panels A and B?
- 1) Center the label "Concentration (μM)" under the middle panel of Fig 2A; remove the label under the left-most panel. The same comment applies to Fig. 6B.
- 2) Report the goodness-of-fit values for Fig. 2B. The same comment applies for Supp. Fig. 2B, Supp. Fig. 3. and Supp. Fig. 4B.
- 3) Provide a numbered reference to "a previous study" on line 214.
- 4) Typo on line 288: "mdoeled" -> "modeled".
- 5) On line 563, does "and (15)" refer to a non-existent Eq. (15) or to reference 15? The "and" and "respectively" are confusing.
- 6) How does the estimation of PRC using SR parameters depend on the number of theta samples used to approximate Eq. 7? Line 583 says that 100 samples are used. Could adaptive quadrature be used instead?
- 7) Remove the box around "0" in Supp. Fig. 8A.

Reviewer #2

(Remarks to the Author)

Phase Response Curves (PRCs) are the core method to quantify the synchronization and entrainment of oscillators. Traditional approaches require an enormous effort to deliver pulses at different times, to measure asymptotic phases, etc. Manella et al. (Ref.11) developed an exciting technique to get PRCs from desynchronized cells. They applied their tool also to multiple stimuli in Ref.12. In a series of papers (Refs. 13-15) Winfree's singularity response (SR) is exploited to characterize models, single cells, bulk data, and slices. This series of papers (Refs. 11-15) is an excellent combination of theory and experiments.

The submitted manuscript illustrates with toy models the link between SR and PRCs (Fig.1). The authors fit dose-response curves to Manella data (Fig.2) and predict SR parameters from published data. Then they re-analyze pre-stimulation data (Fig.4). In Figures 5 and 6 own data (cultured cells and slices) are used to estimate SR parameters. The results are interesting, clearly presented, and convincing.

Keeping in mind Refs.11-15 the authors should emphasize the novelty of the current manuscript more clearly. Moreover, some limitations of PRCs should be discussed, and differences between single cells, bulk data, slices, and organisms might

be addressed.

Specific points:

Only for almost circular limit cycles and fast relaxation, PRCs have for small perturbation sinusoidal shapes, a predictable singularity, and a direct link to entrainment phases (see e.g. J. Aschoff, C. Pittendrigh, C.H. Johnson, A. Granada 2013). In mammals, there are relatively slow relaxations to the limit cycle (Czeisler, Comas & Daan), and consequently, the effects of multiple pulses and the entrainment characteristics (range, phase) are more complex. This implies, that PRCs and SR from a few days as in Fig.5a contain limited predictability.

In excitable systems, quite different PRCs are possible (e.g., advance-only PRCs in integrate-and-fire models). Is there SR also helpful?

Abraham et al. 2010 discuss weak and strong oscillators (lung slices versus brain slices) with quite different entrainment properties. The data analyzed in the manuscript seem to contain mainly weak oscillators. A clear distinction between weak and strong oscillators was published by Aschoff and Pohl 1978. For strong oscillators (e.g., mammals with light pulses) the singularity might be difficult to achieve with single pulses (Czeisler 1991).

Minor points:

Figure 1A and B are confusing to me. In A $\Phi=0$ and B $\Phi=0.25$ is used.

In Figure 2 different methods (regression ...) might be compared to get the PRC from the noisy data. I find the number of digits in the regressions in A and B not useful.

I find „background effects“ a bit misleading. Ist more about previous stimuli mimicking different GC levels.

Line 386: MF is explained only later in Fig.6.

Version 1:

Reviewer comments:

Reviewer #1

(Remarks to the Author)

The authors have put their present work in better context satisfying my primary concern. I am also satisfied with their responses to my minor comments.

I have several comments about the newly added Supplementary Fig. 7. This was added in response to reviewer 2's inquiry about how the SR might perform with different types of PRCs. I find Supplementary Fig. 7 to be confusing and irrelevant to the current paper. None of the PRCs studied are advance-only and the main text only has a tenuous reference to the figure (on line 210). I would not include this figure, but if the authors insist on including it then I suggest the following. 1) Give details: Include the equations that explicitly give the dynamics of V along with the threshold and reset values. Describe the stimulation. Is the stimulation continual? The behaviors in the figure differ from my expectations for an integrate-and-fire model. 2) Non-dimensionalize. Milliseconds and millivolts are somewhat of a distraction for the point being made. Also, state that phase is uniform between V_{reset} and $V_{threshold}$. 3) Explain why panel c has both delay and advance regions. 4) In neurons, the stimulation would more likely be a kick up (excitatory) rather than the kick down shown. 5) What is happening with the SR curve near input strength = 0 in panel e? 6) Use the same ticks for phase in panels b and c. Maybe transpose panels b,c,d,e so that the common axes can be shared.

Reviewer #2

(Remarks to the Author)

The authors revised the manuscript carefully and convincingly. The extensive supplementary material provides interesting and relevant details.

Responses to reviewer's comments.

We thank all reviewers for their constructive and valuable comments, which have helped us improve our manuscript significantly. We have carefully addressed each comment and revised the manuscript accordingly. The following are the authors' point-by-point responses to the reviewers' comments. We have earnestly tried to address all the concerns raised by the reviewers.

Reviewer #1 (Remarks to the Author):

This article is another in a sequence of articles in which the authors expound the benefits of the singularity response in circadian biology. In this installment, the dose-response is studied. The results show that the singularity response is an effective and efficient method to measure phase-response curves for a range of stimuli strengths. Remarkably, the singularity response is effective for multiple stimuli, as well as in both cultured cells and tissues.

Thank you very much for your positive feedback. It is encouraging.

My only significant suggestion is to include a summary of the prior work in the introduction. Detail the author's prior results on the singularity response and put the new results in this article in a proper context.

We have added to the introduction a summary of our previous study on the singularity response, outlining what we have shown in our prior studies and how this new study builds on those findings. Additionally, we emphasized that previous studies have not sufficiently addressed predicting PRC under various conditions by quantifying different PRCs (lines 46–50 and 66–79).

minor comments:

0) Figure 1 illuminates the approach well. Is there a good reason to use a different input direction $\Phi = 0$ vs $\Phi = 0.25$ in panels A and B?

The reason for setting Φ to 0.25 in Figure 1B was to make it easier to see the positions of the stable and unstable points. To unify the Φ in Figure 1a and 1b, we modified the input direction Φ in Figure 1a to 0.25 (rad/2 π).

1) Center the label "Concentration (μM)" under the middle panel of Fig 2A; remove the label under the left-most panel. The same comment applies to Fig. 6B.

The labels in Fig. 2a, 5b and 6b were unified to μM , respectively, and " $(\times 10^3)$ " was added

to the corresponding panel where the order of concentration was mM.

2) Report the goodness-of-fit values for Fig. 2B. The same comment applies for Supp. Fig. 2B, Supp. Fig. 3. and Supp. Fig. 4B.

We have added the coefficient of determination for each approximation curve and the definition of it (Figs. 2, 5, 6 and new Supp. Fig. 3–5, 9; lines 685–688).

3) Provide a numbered reference to "a previous study" on line 214.

We have added a reference to the corresponding sentence (line 238).

4) Typo on line 288: "mdoeled" -> "modeled".

We have corrected it (line 314).

5) On line 563, does "and (15)" refer to a non-existent Eq. (15) or to reference 15? The "and" and "respectively" are confusing.

This "(15)" refers to reference 15, and "and" was inserted by mistake. This "respectively" refers to "amplitude and phase," but this was unnecessary. Thus, we have also removed this word (lines 655).

6) How does the estimation of PRC using SR parameters depend on the number of theta samples used to approximate Eq. 7? Line 583 says that 100 samples are used. Could adaptive quadrature be used instead?

If the number of theta samples is more than 100, there is little error in the calculation of Eq. 7 (new Supp. Fig. 1). As shown in Figure 1b, when the stimulus is strong, the PRC has discontinuous points, so the adaptive quadrature may not converge the results depending on the calculation method.

Since the error in SR parameters with the number of theta sample is also relevant to the calculation of SR parameters from experimental PRC, we have added an explanation (lines 104–107).

7) Remove the box around "0" in Supp. Fig. 8A.

We have modified the graph. The horizontal axis of new Supp. Fig. 11a and Fig. 5jk were also modified for better appearance.

Reviewer #2 (Remarks to the Author):

Phase Response Curves (PRCs) are the core method to quantify the synchronization and entrainment of oscillators. Traditional approaches require an enormous effort to deliver pulses at different times, to measure asymptotic phases, etc. Manella et al. (Ref.11) developed an exciting technique to get PRCs from desynchronized cells. They applied their tool also to multiple stimuli in Ref.12. In a series of papers (Refs. 13-15) Winfree's singularity response (SR) is exploited to characterize models, single cells, bulk data, and slices. This series of papers (Refs. 11-15) is an excellent combination of theory and experiments.

The submitted manuscript illustrates with toy models the link between SR and PRCs (Fig.1). The authors fit dose-response curves to Manella data (Fig.2) and predict SR parameters from published data. Then they re-analyze pre-stimulation data (Fig.4). In Figures 5 and 6 own data (cultured cells and slices) are used to estimate SR parameters. The results are interesting, clearly presented, and convincing.

Keeping in mind Refs.11-15 the authors should emphasize the novelty of the current manuscript more clearly. Moreover, some limitations of PRCs should be discussed, and differences between single cells, bulk data, slices, and organisms might be addressed.

Thank you for your suggestion. We have added a more detailed explanation in the introduction about the previous literature and what we are addressing in this study as in the comment to the Reviewer 1 (lines 46–50 and 66–79). The limitations of the PRCs estimated in the current study are discussed below.

Specific points:

Only for almost circular limit cycles and fast relaxation, PRCs have for small perturbation sinusoidal shapes, a predictable singularity, and a direct link to entrainment phases (see e.g. J. Aschoff, C. Pittendrigh, C.H. Johnson, A. Granada 2013). In mammals, there are relatively slow relaxations to the limit cycle (Czeisler, Comas & Daan), and consequently, the effects of multiple pulses and the entrainment characteristics (range, phase) are more complex. This implies, that PRCs and SR from a few days as in Fig.5a contain limited predictability.

For the Stuart-Landau equation, which includes a circular limit cycle as shown in Fig. 1a, the SR parameters calculated from the PRC and measured from the population response

agree when the oscillators sufficiently recovered from the deviation from the limit cycle after stimulation (new Supp. Fig. 6). However, if the relaxation to the limit cycle was slow, the SR amplitude was overestimated. In this case, the error can be reduced by normalizing the amplitude by the maximum response as we did in the experimental data to obtain the dose-response curve in Fig. 5b and Fig. 6ac. If the relaxation is extremely slow, as in the harmonic oscillators, the reset amplitude would keep increasing with stronger stimuli, which may be distinguishable from the present results.

The predicted PRC may not directly explain the entrainment with the periodic environment, since the effects of deviations from the limit cycle may become more complex, as you pointed out. However, such effects are likely observed with particularly strong stimuli, so the present results quantifying the stimulus intensity are still important for understanding how the circadian clock synchronizes with various environments.

We have added the results of simulation using Stuart-Landau model (new Supp. Fig. 6) and the above discussion to the manuscript (line 513–527 and Supplementary Text 1).

In excitable systems, quite different PRCs are possible (e.g., advance-only PRCs in integrate-and-fire models). Is there SR also helpful?

From previous studies (8–12), the PRCs in cultured cells and tissues are usually sinusoidal when the stimulus is weak, and linear when the stimulus is strong as shown in Fig. 1b. Therefore, the PRC could be predicted from the SR parameters in this study (Figs. 2 and 3). However, in the integrate-and-fire models, the PRC only shows phase advances, and the shape of the PRC differs significantly from the circular limit cycle model in Fig. 1a (new Supp. Fig. 7). Therefore, even though the SR parameters are the same, corresponding PRCs should be different between the circular limit cycle model and integrate-and-fire model. However, even in this case, SR amplitude showed a change close to the Hill curve. We also confirmed that this SR parameter can be estimated from the response of desynchronized population. Thus, quantification of the SR parameter is possible even when PRC is not symmetrical shape as shown in Fig. 1b. Therefore, for systems other than circadian clocks, the quantification of PRC using SR is still useful, but it would be necessary to assume a shape of PRC corresponding to that system to predict PRC from SR parameters.

We have added the results of simulation using the integrate-and-fire model (new Supp. Fig. 7) and discussion to the manuscript (lines 527–538 and Supplementary Text 1).

Abraham et al. 2010 discuss weak and strong oscillators (lung slices versus brain slices) with quite different entrainment properties. The data analyzed in the manuscript seem to

contain mainly weak oscillators. A clear distinction between weak and strong oscillators was published by Aschoff and Pohl 1978. For strong oscillators (e.g., mammals with light pulses) the singularity might be difficult to achieve with single pulses (Czeisler 1991).

Indeed, circadian rhythms are often not spontaneously desynchronized at the SCN or individual level. However, in our experiment, spontaneous decay was also observed in the SCN circadian rhythm, and SR-like responses to forskolin could be measured (new Supp. Fig. 13). Desynchronization of SCN can be also induced by several treatments, such as tetrodotoxin, an inhibitor of voltage-gated sodium channels, MDL-12330A, an inhibitor of adenylate cyclase, and constant light conditions (9, 21, 31).

Although we found that the dose-response curve of SR amplitude for forskolin in SCN was not Hill curve-like, at least for the single cell, bulk, and peripheral tissue slices, the changes in SR parameters under various conditions can be represented by the same model based on the dose-response curve. Future studies will show whether the SR-based models can express the entrainment properties in SCN and individual rhythms.

We have added these discussions to the manuscript (lines 401–412 and 538–549).

Minor points:

Figure 1A and B are confusing to me. In A $\Phi=0$ and B $\Phi=0.25$ is used.

This is related to a comment from reviewer#1.

The input direction Φ in Figure 1a was modified to 0.25 ($\text{rad}/2\pi$).

In Figure 2 different methods (regression ...) might be compared to get the PRC from the noisy data. I find the number of digits in the regressions in A and B not useful.

We have reduced and aligned the number of digits in the overall Figure. More detailed values are given in the source file.

I find „background effects“ a bit misleading. It's more about previous stimuli mimicking different GC levels.

We have noted that “background” also indicates the level of resetting reagents in medium (line 233).

Line 386: MF is explained only later in Fig.6.

Thank you for your careful reading. “MF” was first mentioned in the explanation of Figure 5 (line 340).

Other revisions:

1. We have corrected the maximum value of input strength in Figure 1b.
2. We have corrected the significance symbols in Figure 5m and 6m.
3. We have corrected the parameters of dose-response curve in kidney (Figure 6c).
4. We have specified that the dose-response curve in WAT was calculated excluding the value of 100 nM in Figure 6c (lines 443–444).
5. We have noted that each individual data indicates the results of independent experiments and experiments were performed three times or more in Figure 5 and 6 (lines 381–383 and 457–458).
6. We have changed labels on figure panels from capital letters to small letters.
7. We have provided a source file for this manuscript, which includes individual data and the exact *p*-value in Figure 5 and 6.

Responses to reviewer's comments.

We thank all reviewers for their constructive and valuable comments, which have helped us improve our manuscript significantly. We have carefully addressed the comments and revised the manuscript accordingly.

Reviewer #1 (Remarks to the Author):

The authors have put their present work in better context satisfying my primary concern. I am also satisfied with their responses to my minor comments.

I have several comments about the newly added Supplementary Fig. 7. This was added in response to reviewer 2's inquiry about how the SR might perform with different types of PRCs. I find Supplementary Fig. 7 to be confusing and irrelevant to the current paper. None of the PRCs studied are advance-only and the main text only has a tenuous reference to the figure (on line 210). I would not include this figure, but if the authors insist on including it then I suggest the following. 1) Give details: Include the equations that explicitly give the dynamics of V along with the threshold and reset values. Describe the stimulation. Is the stimulation continual? The behaviors in the figure differ from my expectations for an integrate-and-fire model. 2) Non-dimensionalize. Milliseconds and millivolts are somewhat of a distraction for the point being made. Also, state that phase is uniform between V_{reset} and $V_{threshold}$. 3) Explain why panel c has both delay and advance regions. 4) In neurons, the stimulation would more likely be a kick up (excitatory) rather than the kick down shown. 5) What is happening with the SR curve near input strength = 0 in panel e? 6) Use the same ticks for phase in panels b and c. Maybe transpose panels b,c,d,e so that the common axes can be shared.

As you pointed out, discussing the PRC of a model of neural firing in a paper on circadian rhythms may confuse the reader. Since the PRC is also measured other than circadian rhythms and neural firing, it would have been more appropriate to discuss the SR for other systems in general terms and not just simulations on one of the neural firing models. Therefore, we have removed Supplementary Fig. 7 and revised the relevant section to a more general discussion (lines 531-538).

Reviewer #2 (Remarks to the Author):

The authors revised the manuscript carefully and convincingly. The extensive supplementary material provides interesting and relevant details.

As we responded to Reviewer #1's comment for Supplementary Fig. 7, although we understand that the discussion of the PRC for other systems than circadian rhythm is important, it should have been more general rather than focusing on specific models. Thus, we revised the relevant section (lines 531-538), removing the Supplementary Fig. 7. The specific discussion for the integrate-and-fire models was included in the response to reviewers in the first revision.